# Class Balance Matters to Active Class-Incremental Learning

## ABSTRACT

Few-Shot Class-Incremental Learning has shown remarkable efficacy in efficient learning new concepts with limited annotations. Nevertheless, the heuristic few-shot annotations may not always cover the most informative samples, which largely restricts the capability of incremental learner. We aim to start from a pool of large-scale unlabeled data and then annotate the most informative samples for incremental learning. Based on this purpose, this paper introduces the Active Class-Incremental Learning (ACIL). The objective of ACIL is to select the most informative samples from the unlabeled pool to effectively train an incremental learner, aiming to maximize the performance of the resulting model. Note that vanilla active learning algorithms suffer from class-imbalanced distribution among annotated samples, which restricts the ability of incremental learning. To achieve both class balance and informativeness in chosen samples, we propose **C**lass-**B**alanced **S**election (**CBS**) strategy. Specifically, we first cluster the features of all unlabeled images into multiple groups. Then for each cluster, we employ greedy selection strategy to ensure that the Gaussian distribution of the sampled features closely matches the Gaussian distribution of all unlabeled features within the cluster. Our CBS can be plugged and played into those CIL methods which are based on pretrained models with prompts tunning technique. Extensive experiments under ACIL protocol across five diverse datasets demonstrate that CBS outperforms both random selection and other SOTA active learning approaches.

## CCS CONCEPTS

• **Computing methodologies** → **Active learning settings**; **Lifelong machine learning**.

## KEYWORDS

class-incremental learning, few-shot class-incremental learning, active learning

## 1 INTRODUCTION

Few-shot class-Incremental Learning (FSCIL) aims to learn new classes with few-shot data without catastrophic forgetting of the preceding learned knowledge. Compared to standard class incremental learning (CIL) [69] which needs extensive labeled training data per session, FSCIL significantly reduces the cost of obtaining labeled samples, gaining wide attention and notable advances within the incremental learning field [47, 49, 66].

*Conference acronym 'ACM MM', June 03–05, 2018, Woodstock, NY*
© 2024 Copyright held by the owner/author(s). Publication rights licensed to ACM.
ACM ISBN 978-1-4503-XXXX-X/18/06
https://doi.org/XXXXXXX.XXXXXXX

Nevertheless, the process of few-shot labeling is usually heuristic, since in FSCIL scenarios, the annotated candidates are usually random selected and are seldom chosen by specific rules. Therefore, the quality of annotated samples may largely varies among different candidates, thus wasting the merits from efficient annotation procedure. Instead of vanilla few-shot labeling, gathering a large amount of unlabeled data is relatively easy and cheap, and such data can precisely represent the distribution of corresponding categories in realistic world. Given this scenario, in each incremental session, one has the opportunity to tap into a large pool of unlabeled data, selecting only a handful for labeling and subsequent training of an incremental learner. This strategy has the similar cost with FSCIL, but is more reasonable and effective in incremental learning scenarios. Our aim is to select the most informative samples that can significantly enhance the learner's performance to its highest potential.

In this paper, we present **Active Class-Incremental Learning (ACIL)** task. The most significant distinction between ACIL and FSCIL lies in their approach to forming training sets in each incremental session. Specifically, the protocol of FSCIL randomly selects an equal number of samples for each class in each incremental session, which ensures a class-balanced training set for training the incremental learner. The balanced training set in each new task ensures the remarkable performance in incremental learning scenarios. In contrast, achieving such class-balanced sampling from an unlabeled pool in ACIL task presents a significant challenge. Our empirical results for adopting the advanced FSCIL method in ACIL scenarios reveal that random selection of samples from this unlabeled pool often leads to severe class imbalance within each incremental session, as illustrated in Fig. 1 (a). Using such a class-imbalanced training set will harm the performance of an incremental learner, as shown in Fig. 1 (g). Moreover, we further find that applying existing active learning methods [6, 23, 41, 44] to select samples also fails to effectively obtain a class-balanced training set, even worse than that of random sampling, as shown in Fig. 1 (b) to Fig. 1 (e). Consequently, the integration of these active learning methods within the ACIL framework tends to degrade performance even further when compared to random selection of samples, as shown in Fig. 1 (g). These observations have motivated us to design such a more balanced active selection algorithm for ACIL that leading to efficient yet effective incremental learning.

To this end, we propose **C**lass-**B**alanced **S**election (**CBS**) approach for Active Class-Incremental Learning, which considers both the class balance and informativeness of the selected samples to benefit the training procedure of the incremental learner. The key idea of our CBS is to *ensure the distribution of selected samples closely mirrors the distribution of the entire unlabeled pool,* thereby achieving a class-balanced selection while also selecting samples that are representative and diverse. Specifically, at the beginning of an incremental session, all unlabeled data are fed into the pretrained feature extractor to obtain corresponding features. These features are divided into multiple clusters, and then we attempt to select

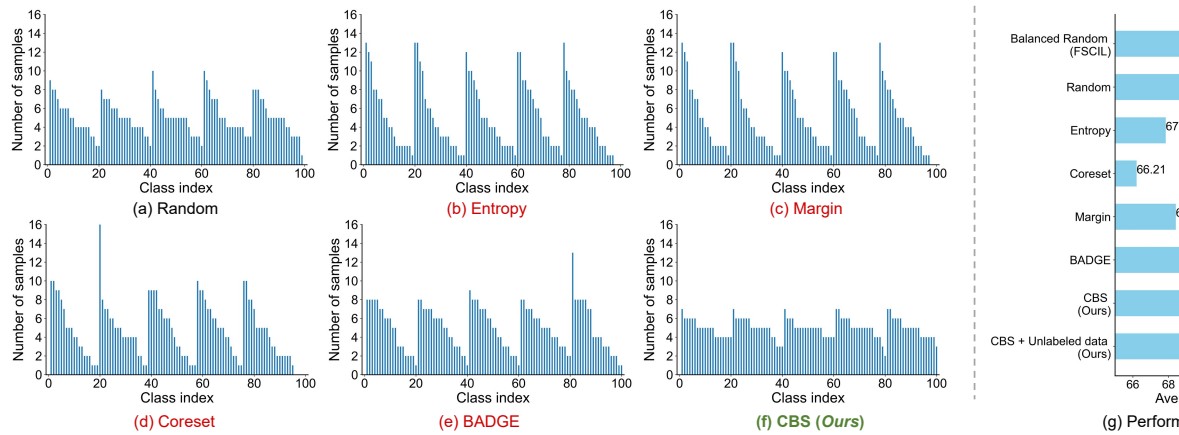

**Figure 1: Analysis of applying various active learning approaches to LP-DiF [25] on CUB-200 under ACIL protocol (see Sec. 4.1). (a) to (f) show the the class distribution (first 100 classes of CUB-200) of samples selected by different active learning approaches and (g) compares their corresponding performance on the test set. Clearly, the samples selected by existing active learning methods (*i.e.,* (b) to (e)) exhibit more severe class imbalance compared to random selection (*i.e.,* (a)), which leads to that their corresponding performance is worse than random sampling. However, our proposed CBS (*i.e.,* (f)) can achieve more class-balanced sampling, thereby outperforming both random sampling and existing active learning methods.**

samples from each cluster. For each cluster, we design a greedy selection method that aims to ensure the distribution of the selected features closely approximates the distribution of all features in this cluster. Finally, the samples selected from each group are collected to form the final selection, which are then annotated by oracle human-based annotation and used to train the incremental learner.

Our CBS can be plug-and-played into the recently proposed CIL or FSCIL methods which are built on pretrained models [16, 38] with employing prompts tunning technique [27, 71] to learn new knowledge, *e.g.*, L2P [54], DualPrompt [53] and LP-DiF [25]. Particularly, when applying CBS to LP-DiF, we further exploit the unlabeled data not selected by CBS to improve the estimation method for the feature-level Gaussian distribution, which can generate higher-quality pseudo features for knowledge replay to enhance the model's resistance to catastrophic forgetting. Experimentally, applying our proposed methods to LP-DiF outperforms existing active learning methods and random selection, as shown in Fig. 1 (g).

Our contributions in this paper are summarized as follows:

1) We present Active Class-Incremental Learning task and empirically reveal that class-balanced annotations are crucial for promising incremental learning.

2) We propose a model-agnostic approach namely **C**lass-**B**alanced **S**election (**CBS**), which considers both the class balance and informativeness of the selected samples for benefiting training the incremental learner. To achieve the such sampling ability mentioned above, CBS ensures that the distribution of the selected samples is as close as possible to the distribution of samples in the entire unlabeled pool by a designed greedy selection method.

3) We incorporate CBS into L2P, DualPrompt and LP-DiF, which represent CIL methods based on pretrained models with employing prompt tunning technique. Extensive evaluations and comparisons on five datasets show the effectiveness of

CBS in ACIL, and surpasses existing SOTA active learning methods and random selection.

## 2 RELATED WORK

### 2.1 Class-Incremental Learning

Class-incremental learning (CIL) [69] addresses the challenge of adapting models to recognize new classes over time without forgetting previously learned knowledge, enabling continuous model evolution in dynamic environments. To date, a significant body of work has addressed CIL problem, encompassing several families: data replay [5, 9, 10, 26], knowledge distillation [19, 24, 33, 40], parameters regularization [29, 32, 61, 64], and dynamic networks [4, 43, 56, 63]. Recently some works [17, 46, 48, 51–54, 59] employ prompt tuning techniques on pretrained model (*e.g.*, ViT [16]) to capture new knowledge and preserve old knowledge by learning different prompts.

Although CIL has received widespread attention and development, the need for extensive labeled data in each session raises concerns about the cost of annotation. In this paper, we introduce Active Class-Incremental Learning (ACIL), where for each session, only a number of unlabeled data can be obtained. The model selects a small number of valuable samples to return to humans for annotation, significantly reducing the cost of labeling.

### 2.2 Few-Shot Class-Incremental Learning

The objective of few-shot class incremental learning approaches [49] (FSCIL) is to facilitate the model's training in adopting new classes incrementally, leveraging merely a sparse set of data for each incremental session. Current research in the field can be systematically organized into four distinct categories: replay-based methods [11, 15, 25, 31], meta-learning-based methods [13, 22, 35, 60, 72, 74], dynamic network-based methods [20, 47, 57, 58] and feature space-based methods [1–3, 12, 28, 67, 68, 70, 73]. Recently, Huang *et al.* [25]

proposes LP-DiF, which utilizes prompt tuning based on CLIP to learn new knowledge and estimates a Gaussian distribution at the feature level to facilitate the replay of old knowledge. All these methods assume that only a small amount of data can be acquired in each session. While in this paper, we believe that a large amount of unlabeled data can be obtained in each session under a lower cost. Then we design an active learning approach to select the most valuable samples to label. Compared to FSCIL, we aim to achieve the highest possible model performance without increasing the annotation cost.

### 2.3 Active Learning.

**Active Learning for Image Classification.** Active Learning for image classification [7, 18, 23, 37, 41, 45, 55] aims to efficiently utilize a limited label budget by selecting the most valuable samples for labeling to maximize the performance of a model. Traditional AL strategies, such as Margin [41], Entropy [23], and DBAL [18], focus on uncertainty sampling, where samples for which the model has the highest uncertainty are prioritized. While GEAL [55] and Coreset [45] emphasize strategies that ensure a diverse set of samples is selected. In the realms of the low-budget regime, Typiclust [21] and ProbCover [62] are proposed to select the typical samples which have highest density in the representation space. Recently, BADGE [7] explores hybrid methodologies that integrate aspects of uncertainty and diversity to harmonize the advantages of each strategy.

**Active Learning for Class-Incremental Learning.** Currently, there is little work exploring the application of active learning in class-incremental learning. Ayub *et al.* [8] introduces the active sampling approach to the task of scene recognition with a real humanoid robot. However, we are the first to study active class-incremental learning aimed at a more general image classification problem, and we find that the samples selected by existing active learning methods exhibit class imbalance, leading to sub-optimal performance of class-incremental learners. Furthermore, this paper designs a class-balanced sampling method to improve the performance of the model.

## 3 PROPOSED METHOD

**Problem Formulation.** Referencing the problem formulations of Class-Incremental Learning (CIL) [69] and Active Learning (AL) [65], we first formulate the problem setting of ACIL. The purpose of ACIL is to select informative samples from a pool of unlabeled images provided by a designed active selection algorithm in each session, which are then annotated by humans to train a class-incremental model, ensuring the model learns new categories without forgetting previously acquired knowledge. Formally, an incremental learner can obtain a sequence of unlabeled pools $[\mathcal{D}^1_{\text{Pool}}, \mathcal{D}^2_{\text{Pool}}, \ldots, \mathcal{D}^T_{\text{Pool}}]$ over $T$ incremental sessions, where $\mathcal{D}^t_{\text{Pool}}$ denotes the unlabeled pool of session $t$, containing $N^t$ unlabeled images $\{\mathbf{x}_i\}^{N^t}_{i=1}, \forall \mathbf{x}_i \in \mathbb{R}^{H \times W \times 3}$. Let $C^t$ be the class space to which data in $\mathcal{D}^t_{\text{Pool}}$ may belong. Following the setting of CIL [69], for different sessions, the class spaces are non-overlapping, *i.e.* $\forall t_1, t_2 \in \{1, 2, \ldots, T\}$ and $t_1 \neq t_2, C^{t_1} \cap C^{t_2} = \varnothing$. In incremental session $t$, $B$ (*i.e.*, the labeling budget, $B < N^t$) images are selected from $\mathcal{D}^t_{\text{Pool}}$ by a designed active selection algorithm, and then the labels for these images are

---

**Algorithm 1:** Active Class-Incremental Learning

**Input:** The number of sessions $T$; a sequence of unlabeled pools $[\mathcal{D}^t_{\text{Pool}}]^T_{t=1}$; class space in each session $[C^t]^T_{t=1}$; labeling budget of each session $B$; pretrained model $f(\cdot|\Theta^0)$ with randomly initialized learnable parameters $\Theta^0$ (*e.g.*. prompts); CIL method $\mathcal{A}(\cdot)$ (*e.g.*. L2P, DualPrompt or LP-DiF); oracle $\phi(\cdot)$

**Output:** Model $f(\cdot|\Theta^T)$ with optimized parameters $\Theta^T$.

1   $\mathcal{M}^0 \leftarrow \emptyset$;   *// initialize the memory buffer, which is used to store the Gaussian distributions.*

2   $E(\cdot)$ denotes the pretrained feature extractor of $f(\cdot|\Theta^0)$;

3   **for** *each session* $t \in \{1, 2, \ldots, T\}$ **do**

4     $\mathcal{S}^t \leftarrow$ ClassBalancedSelection $(\mathcal{D}^t_{\text{Pool}}, |C^t|, B, E(\cdot))$;   *// Call Alg. 2 to select samples.*

5     $\mathcal{D}^t_{\text{Labeled}} \leftarrow \phi(\mathcal{S}^t)$;   *// Obtain labels from the oracle.*

6     $\Theta^t \leftarrow \mathcal{A}(\mathcal{D}^t_{\text{Labeled}}, f(\cdot|\Theta^{t-1}), \mathcal{M}^{t-1})$;   *// Using $\mathcal{D}^t_{\text{Labeled}}$ and $\mathcal{M}^{t-1}$ to train the $\Theta^{t-1}$ to $\Theta^t$;.*

7     $\mathcal{R}^t \leftarrow \mathcal{D}^t_{\text{Pool}} \setminus \mathcal{S}^t$;   *// The set of unlabeled data not be selected.*

8     $\mathcal{M}^t \leftarrow$ DistributionEstimation $(\mathcal{D}^t_{\text{Labeled}}, \mathcal{R}^t, C^t, f(\cdot|\Theta^t))$;   *// Call Alg. 3 to select samples to estimate the Gaussian distributions.*

9     $\mathcal{M}^t \leftarrow \mathcal{M}^{t-1} \cup \mathcal{M}^t$;   *// Update the memory buffer.*

10   **end**

11   Return $f(\cdot|\Theta^T)$

---

obtained from an oracle $\phi(\cdot)$ (*i.e.*, human annotations), forming a labeled set $\mathcal{D}^t_{\text{Labeled}} = \{(\mathbf{x}_i, y_i)\}^B_{i=1}$, where $y_i \in C^t$. Then, the incremental learner is trained on $\mathcal{D}^t_{\text{Labeled}}$ with an optional small memory buffer $\mathcal{M}$ which is used to store the old knowledge (*e.g.*, exemplars). After training, the incremental learner is evaluated on a test set $\mathcal{D}^t_{\text{Test}}$, the class space of which is union of all the classes encountered so far, *i.e.*, $C^1 \cup C^2 \cdots \cup C^t$, to assess its performance on both new and old classes.

### 3.1 Approach Overview

To tackle ACIL task efficiently and effectively, we aim to design such a active selection method for ACIL, that it should not only be able to select informative samples, but also ensure that the selected samples have good class balance. To this end, we propose Class-Balance Selection (CBS) strategy that considers both the class balance and informativeness of the selected samples. The key idea of our CBS is to ensure that the distribution of selected samples closely mirrors the distribution of unlabeled data from corresponding categories, thereby achieving a class-balanced selection while ensuring their representativeness. The merit of CBS is that it can be plug-and-played into state-of-the-art CIL or FSCIL methods with pretrained models [16, 38] and employ prompts tunning technique [27, 71] to learn new knowledge, *e.g.*, L2P [54], DualPrompt [53] and LP-DiF [25].

The whole pipeline to address the ACIL problem is illustrated in Alg. 1, where the blue pseudo code is specifically only for application to LP-DiF. Generally, at the beginning of session $t$, we first select a set of samples $\mathcal{S}^t$ from the given unlabeled pool $\mathcal{D}^t_{\text{Pool}}$ by proposed CBS, which is detailed in Alg. 2, Then, these selected samples will be labeled by the oracle $\phi(\cdot)$, obtaining the labeled set $\mathcal{D}^t_{\text{Labeled}}$. After that, the incremental learner is trained on $\mathcal{D}^t_{\text{Labeled}}$

---

**Algorithm 2:** ClassBalancedSelection

**Input:** Unlabeled pool $\mathcal{D}_{\text{Pool}}^t$; the number of classes in this session $|C^t|$; budget size $B$; pretrained feature extractor $E(\cdot)$;

**Output:** A set of selected samples $\mathcal{S}^t$;

1  $\mathcal{S}^t \leftarrow \emptyset$;  // Initialize the selected set.

2  $N^t \leftarrow |\mathcal{D}_{\text{Pool}}^t|$;

3  $\mathcal{F}^t = \{\mathbf{f}|\mathbf{f} = E(\mathbf{x}) \wedge \mathbf{x} \in \mathcal{D}_{\text{Pool}}^t\}$;  // Use feature extractor to extract image feature for each unlabeled image.

4  Cluster $\mathcal{F}^t$ into $\mathcal{G} = \{\mathbf{G}_1, \mathbf{G}_2, \ldots, \mathbf{G}_{|C^t|}\}$ by K-means;

5  **for** each cluster $j \in \{1, 2, \ldots, |C^t|\}$ **do**

6     $M_j \leftarrow |\mathbf{G}_j|$;

7     $K_j \leftarrow \lceil M_j \times \frac{B}{N^t} \rceil$;  // The number of samples to select for this cluster.

8     $\mathcal{N}(\boldsymbol{\mu}_j, \boldsymbol{\sigma}_j^2) \leftarrow P(\mathbf{G}_j)$;  // Estimate the Gaussian distribution of the entire cluster.

9     $\mathbf{S}_j \leftarrow \{\mathbf{f}_{\text{selected}} : \arg\min_{\mathbf{f} \in \mathbf{G}_j} ||\mathbf{f} - \boldsymbol{\mu}_j||\}$;  // Select the sample closest to the mean vector as the first chosen sample.

10    **for** select $k \in \{2, \ldots, K_j\}$-th samples **do**

11       $\mathbf{f}_{\text{selected}} \leftarrow \arg\min_{\mathbf{f} \in \mathbf{G}_j \backslash \mathbf{S}_j} D_{\text{KL}}(\mathcal{N}(\boldsymbol{\mu}_j, \boldsymbol{\sigma}_j^2)|P(\mathbf{S}_j \cup \{\mathbf{f}\}))$;  // Select such a sample, that adding this sample to the selected set minimizes the KL divergence between the distribution of the selected set and the distribution of the entire cluster.

12       $\mathbf{S}_j \leftarrow \mathbf{S}_j \cup \{\mathbf{f}_{\text{selected}}\}$;

13    **end**

14    $\mathcal{S}^t \leftarrow \mathcal{S}^t \cup \{\mathbf{x}|\mathbf{f} = E(\mathbf{x}) \wedge \mathbf{f} \in \mathbf{S}_j\}$;  // Collect the samples selected in this cluster.

15 **end**

16 Randomly discard $|\mathcal{S}^t| - B$ samples from $\mathcal{S}^t$;

17 Return $\mathcal{S}^t$

---

by using a specific CIL method $\mathcal{A}$. Finally, we finish session $t$ and step into session $t + 1$. In particular, when $\mathcal{A}$ is implemented by LP-DiF, we estimate extra Gaussian distributions for each class, which will be used for generating pseudo features to train the incremental learner [25]. The relevant pseudo code is shown in blue in Alg. 1, and the method for estimating Gaussian distributions is detailed in Alg. 3.

### 3.2 Class-Balanced Selection Strategy

To conduct class-balanced sampling thus ensuring the selected samples precisely match corresponding distribution of original unlabeled data, inspired by active learning methods, we propose Class-Balanced Selection (CBS). In general, CBS consists of two steps, *i.e.*, clustering step and selection step. In clustering step, we first utilize a fix and pretrained feature extractor $E(\cdot)$ to extract features for each image in the unlabeled pool. Notice that the feature extractor has been pre-trained with a large amount of data (*e.g.*, supervised pretraining for ViT in L2P and DualPrompt, contrastive pretraining for CLIP in LP-DiF), therefore the image features it extracts present strong semantic representation capabilities. Then, we use the k-means algorithm [34] to cluster these features into multiple clusters, achieving a coarse classification of these unlabeled samples. In selection step, we select multiple samples from each cluster respectively. For each cluster, we propose a greedy selecting method which efficiently ensures that the distribution of the selected samples is as close as possible to the distribution of

---

**Algorithm 3:** DistributionEstimation

**Input:** Labeled set $\mathcal{D}_{\text{Labeled}}^t$; unlabeled set $\mathcal{R}^t$; class space $C^t$; model $f(\cdot|\Theta^t)$.

**Output:** A set of estimated Gaussian distributions $\mathcal{M}^t$.

1  $\mathcal{M}^t \leftarrow \emptyset$;

2  Using $f(\cdot|\Theta^t)$ to generate pseudo labels for data in $\mathcal{R}^t$ by Eqn. 3, obtaining $\mathcal{D}_{\text{Pseudo}}^t$;

3  **for** each class $c \in C^t$ **do**

4     $\mathcal{D}_c^t \leftarrow \{(\mathbf{x}_i, y_i) \in \mathcal{D}_{\text{Labeled}}^t | y_i = c\} \cup \{(\mathbf{x}_j, \tilde{y}_j) \in \mathcal{D}_{\text{Pseudo}}^t | \tilde{y}_j = c\}$;  // The set of samples with label $c$ or pseudo label $c$.

5     $\mathcal{F}_c^t = \{\mathbf{f}|\mathbf{f} = E(\mathbf{x}) \wedge \mathbf{x} \in \mathcal{D}_c^t\}$;  // $E(\cdot)$ in the feature extractor of $f(\cdot|\Theta^t)$.

6     $\mathcal{N}(\boldsymbol{\mu}_c, \boldsymbol{\sigma}_c^2) \leftarrow P(\mathcal{F}_c^t)$;  // Estimate the Gaussian distribution of class $c$.

7     $\mathcal{M}^t \leftarrow \mathcal{M}^t \cup \{\mathcal{N}(\boldsymbol{\mu}_c, \boldsymbol{\sigma}_c^2)\}$;

8  **end**

9  Return $\mathcal{M}^t$

---

all unlabeled samples within the clusters at feature-level. Finally, the samples selected from each cluster are collected to form the final selection set Since the distribution of selected samples in each cluster is closed to the distribution of all samples in that cluster, the distribution of the final selected samples is close to the distribution of the entire unlabeled pool, which achieves class-balanced sampling while ensuring the representativeness and diversity of the sampled samples. The details of CBS is shown in Alg. 2 and we will highlight the key steps as follows.

**Clustering step.** At the beginning of session $t$, all the unlabeled images $\{\mathbf{x}_i\}_{i=1}^{N^t}$ of $\mathcal{D}_{\text{Pool}}^t$ are fed into the image encoder (*e.g.*, ViT [16]), obtaining their $L_2$-normalized features $\mathcal{F}^t = \{\mathbf{f}_i\}_{i=1}^{N^t}, \mathbf{f}_i \in \mathbb{R}^D$, where $D$ represents the dimension of feature. Then, these features are clustered by K-means algorithm into $|C^t|$ clusters $\mathcal{G} = \{\mathbf{G}_1, \mathbf{G}_2, \ldots, \mathbf{G}_{|C^t|}\}$, where $\mathbf{G}_j = \{\mathbf{x}_i\}_{i=1}^{M_j}$. $M_j$ represents the size of $\mathbf{G}_j$, satisfying $\sum_{j=1}^{|C^t|} M_j = N^t$. Then, we will select samples from each cluster respectively using the following proposed greedy selection algorithm.

**Selection step.** The objective of the greedy selection algorithm is to ensure that the distribution of the selected samples in one cluster is as close as possible to the distribution of all samples within the entire cluster. Motivated by previous FSCIL approaches [25, 59], we use multivariate Gaussian distributions to characterize the samples within each cluster. Formally, let $\mathcal{N}(\boldsymbol{\mu}_j, \boldsymbol{\sigma}_j^2)$ denotes the estimated distribution of $\mathbf{G}_j$. $\boldsymbol{\mu}_j = \frac{1}{M_j} \sum_{i=1}^{M_j} \mathbf{f}_i$ denotes the mean vector; $\boldsymbol{\sigma}_j^2 \in \mathbb{R}^D$ denotes the diagonal values of the covariance matrix, estimated by $\sigma_{jd}^2 = \frac{1}{M_j} \sum_{i=1}^{M_j} (f_{id} - \mu_{jd})^2$, where $\sigma_{jd}^2$ is the $d$-th value of $\boldsymbol{\sigma}_j^2$ and $\mu_{jd}$ is the $d$-th value of $\boldsymbol{\mu}_j$. For a concise representation, we use $P(\cdot)$ to denote the function which estimates the Gaussian distribution $\mathcal{N}(\boldsymbol{\mu}_j, \boldsymbol{\sigma}_j^2)$ from a set $\mathbf{G}_j$, *i.e.*, $\mathcal{N}(\boldsymbol{\mu}_j, \boldsymbol{\sigma}_j^2) \leftarrow P(\mathbf{G}_j)$. Let $\mathbf{S}_j = \{\mathbf{f}_i\}_{i=1}^{K_j}$ be the set of selected samples and $\mathcal{N}(\hat{\boldsymbol{\mu}}_j, \hat{\boldsymbol{\sigma}}_j^2) \leftarrow P(\mathbf{S}_j)$ denotes the corresponding estimated Gaussian distribution, where $K_j$ is the number of selected samples. Practically, $K_j$ can be set by:

$$K_j = \lceil M_j \times \frac{B}{N^t} \rceil, \tag{1}$$

where $\lceil \cdot \rceil$ represents the rounding up operation. We aim to find an optimized $\mathbf{S}_j$ such that $\mathcal{N}(\hat{\boldsymbol{\mu}}_j, \hat{\boldsymbol{\sigma}}_j^2)$ can be as closed as possible to $\mathcal{N}(\boldsymbol{\mu}_j, \boldsymbol{\sigma}_j^2)$, which can be formulated by minimizing the distance between above two Gaussian distributions via Kullback-Leibler (KL) divergence:

$$D_{\mathrm{KL}}(\mathcal{N}(\boldsymbol{\mu}_j, \boldsymbol{\sigma}_j^2)|\mathcal{N}(\hat{\boldsymbol{\mu}}_j, \hat{\boldsymbol{\sigma}}_j^2)) = \frac{1}{2}\sum_{d=1}^{D}\left(\frac{\sigma_{j_d}^2}{\hat{\sigma}_{j_d}^2} + \frac{(\hat{\mu}_{j_d} - \mu_{j_d})^2}{\hat{\sigma}_{j_d}^2} + \ln\left(\frac{\hat{\sigma}_{j_d}^2}{\sigma_{j_d}^2}\right) - 1\right).$$
(2)

Intuitively, we can exhaust all selection schemes and calculate the corresponding $D_{\mathrm{KL}}$, and then select the group of choices that minimizes $D_{\mathrm{KL}}$ as the final selection scheme for this cluster. However, this is a combinatorial problem with $C(M_j, K_j) = \frac{M_j!}{K_j!(M_j-K_j)!}$ possible combinations. As $M_j$ and $K_j$ grow, the number of combinations can increase very rapidly, leading to an explosion in terms of computational cost. To achieve a more efficient selection, we propose a greedy selection algorithm. The key steps of the greedy selection algorithm are shown in lines 9 to line 13 of Alg. 2. We first select the sample that is closest to the $\boldsymbol{\mu}_j$. Next, we respectively add each of the remaining samples to the already selected sample set, calculating the corresponding $D_{\mathrm{KL}}$. The sample that results in the smallest $D_{\mathrm{KL}}$ will be finally chosen. Then, we repeat this process until the number of selected samples reaches $K_j$.

Finally, the samples selected from each cluster are collected to form the final selection set $\mathcal{S}^t = \bigcup_{j=1}^{|C^t|} \mathbf{S}_j$ of session $t$. Considering that Eqn. 1 involves rounding up to determine $K_j$, which could result in $|\mathcal{S}^t|$ may exceed the specified the labeling budget $B$, we randomly discard the excess part, *i.e.*, randomly discard $|\mathcal{S}^t| - B$ samples from $\mathcal{S}^t$.

## 3.3 Incorporate CBS into CIL methods.

In this section, we will introduce how to incorporate CBS with existing state-of-the-art CIL methods to achieve promising performance efficiently. Generally, CBS can be plug-and-played into state-of-the-art CIL methods which are built on pretrained models [16, 38] and employ prompt tuning techniques [27, 71]. We incorporate CBS into several representative works, *i.e.*, L2P [54], DualPrompt [53] and LP-DiF [25] to build the whole ACIL pipeline.

**Incorporate CBS into L2P and DualPrompt.** These two approaches are based on a pretrained ViT [16] and employ visual prompt tunning [27] to encode knowledge from different sessions. We use their pretrained ViT as the feature extractor $E(\cdot)$ to extract image features for the unlabeled data in each session, which is involved in Alg. 2. And then we follow corresponding training procedures to optimize the incremental learner.

**Incorporate CBS into LP-DiF.** LP-DiF is built on CLIP and employ text prompt tuning [71] to new knowledge, and propose to estimate Gaussian distributions for encountered classes, which are used to sample pseudo features to train the prompts in subsequent sessions to prevent from forgetting. We use the pretrained image encoder of CLIP as feature extractor $E(\cdot)$ to extract image features for the unlabeled data. In addition, we further exploit the unlabeled data not selected by CBS to improve the estimation method for the feature-level Gaussian distribution proposed by it, which can generate pseudo features with higher quality for knowledge replay. Formally, let $\mathcal{R}^t = \mathcal{D}_{\mathrm{Pool}}^t \backslash \mathcal{S}^t = \{\mathbf{x}_i\}_{i=1}^{R^t}$ denotes the set of unlabeled

data not selected by CBS, where $R^t = N^t - B$ denotes its size. We use the incremental learner which has trained on $\mathbf{D}_{\mathrm{Labeled}}^t$ to generate pseudo labels for unlabeled samples, forming the pseudo set $\mathcal{D}_{\mathrm{Pseudo}}^t = \{(\mathbf{x}_i, \tilde{y}_i)\}_{i=1}^{R^t}$, where $\tilde{y}_i$ is obtained by:

$$\tilde{y}_i = \arg\max_{c \in C^t} \frac{\exp(\langle \mathbf{f}_i, \mathbf{g}_c \rangle / \tau)}{\sum_{j \in C^t} \exp(\langle \mathbf{f}_i, \mathbf{g}_j \rangle / \tau)},$$
(3)

where $\mathbf{f}_i$ represents the feature of unlabeled image $\mathbf{x}_i$, $\mathbf{g}_j$ represents the text embedding corresponding to class $j$, $\langle \cdot, \cdot \rangle$ represents the cosine similarity of the two features and $\tau$ is the temperature term. Then for each $c \in C^t$, we estimate the Gaussian distribution $\mathcal{N}(\boldsymbol{\mu}_c, \boldsymbol{\sigma}_c^2)$ by the data in $\mathcal{D}_c^t = \{(\mathbf{x}_i, y_i) \in \mathcal{D}_{\mathrm{Labeled}}^t | y_i = c\} \cup \{(\mathbf{x}_j, \tilde{y}_j) \in \mathcal{D}_{\mathrm{Pseudo}}^t | \tilde{y}_j = c\}$. Now, the knowledge of each class in session $t$ is modeled as a Gaussian distribution by both the labeled data and unlabeled data. In subsequent sessions, the previously learned Gaussian distributions are leveraged to sample pseudo-features, combined with the accessible real labeled data to jointly tune the prompt [25]. The relevant pseudo code is shown in blue in Alg. 1, and the method for estimating Gaussian distributions is detailed in Alg. 3.

## 4 EXPERIMENTS

## 4.1 Experiments Setup

**Datasets.** We conduct experiments on selected five publicly available image classification datasets, *i.e.*, CUB-200 [50], CIFAR-100 [30], *mini*-ImageNet [42], DTD [14] and Flowers102 [36], to evaluation our CBS. The first three datasets are commonly utilized for evaluation in CIL or FSCIL, while the latter two datasets are more challenging classification datasets usually adopted to evaluate for vision-language model [38]. We evenly divide each dataset into multiple subsets to construct incremental sessions, and the details are present in the supplementary materials. In addition, we also evaluate the effect of CBS on datasets that the unlabeled pool are inherently class-imbalanced (*e.g.*, CIFAR-100-LT) in the supplementary materials.

**Metrics.** Following existing CIL methods [51, 69], we employ the Avg., which is the average accuracy over each session, as primary metric for performance comparison.

**Class-incremental learning methods.** As mentioned in Sec. 3.3, we incorporate proposed CBS and compared existing SOTA active learning methods into three CIL methods, *i.e.*, 1) L2P [54], 2) Dual-Prompt [53] and 3) LP-DiF [25].

**Active learning methods.** To validate the effectiveness of CBS, we apply seven famous active learning methods on each CIL method for comparison, including 1) Uncertainty-based approaches, *i.e.*, Entropy [23] and Margin [41], which focus on maximizing learning from the model's perspective of uncertainty.; 2) Diversity-based approach, *i.e.*, Coreset [45], which is dedicated to selecting samples with higher diversity; 3) density based-approaches, *i.e.*, Prob-Cover [62] and Typiclust [21], which aim to select the typical samples with highest density in the representation space; 4) hybrid methodology, *i.e.*, BADGE [7], which takes into account both the uncertainty and diversity of the sampled examples. In addition, we conduct random selection for each CIL method, *i.e.*, randomly selecting a batch of samples to label in each session, as the *baseline* method to evaluate each existing active learning method and our

Table 1: Comparison of our method with other active learning approaches when applying them to three CIL methods on five datasets, under $B = 100$. "Avg" represents the average accuracy across all incremental session and "Mean Avg" represents the mean Avg across five datasets. ↑ and ↓ indicate increments and decrements compared with Random selection (*baseline*).

| Methods | Avg ↑ | | | | | Mean Avg ↑ |
|---|---|---|---|---|---|---|
| | CUB-200 | CIFAR-100 | *mini*-ImageNet | DTD | Flowers102 | |
| L2P [54] | | | | | | |
|   + Random (*Baseline*) | 72.26(0.00) - | 66.48(0.00) - | 91.27(0.00) - | 63.18(0.00) - | 97.76(0.00) - | 78.19(0.00) - |
|   + Entropy [23] | 68.37(3.89)↓ | 65.99(0.49)↓ | 88.33(2.94)↓ | 59.65(3.53)↓ | 97.20(0.56)↓ | 75.90(2.29)↓ |
|   + Margin [41] | 70.97(1.39)↓ | 68.67(2.19)↑ | 91.18(0.09)↓ | 63.37(0.19)↑ | 97.73(0.03)↓ | 78.38(0.18)↑ |
|   + Coreset [45] | 61.77(10.49)↓ | 66.00(0.48)↓ | 89.47(1.80)↓ | 56.78(6.40)↓ | 97.62(0.14)↓ | 74.32(3.87)↓ |
|   + BADGE [7] | 72.95(0.69)↑ | 67.80(1.32)↑ | 93.05(1.78)↑ | 64.71(1.53)↑ | 98.79(1.03)↑ | 79.46(1.27)↑ |
|   + Typiclust [21] | 73.07(0.81)↑ | 71.20(4.72)↑ | **93.25**(1.98)↑ | 66.37(3.19)↑ | 98.65(0.89)↑ | 80.50(2.31)↑ |
|   + ProbCover [62] | 68.01(4.25)↓ | 59.67(6.81)↓ | 92.50(1.23)↑ | 52.43(10.84)↓ | 95.73(2.03)↓ | 73.66(4.56)↓ |
|   + DropQuery [39] | 71.23(1.03)↓ | 71.89(5.41)↑ | 91.22(0.05)↓ | 64.56(1.38)↑ | 98.54(0.78)↑ | 79.48(1.29)↑ |
|   **+ CBS (*Ours*)** | **73.96**(1.70)↑ | **72.47**(5.99)↑ | 92.88(1.61)↑ | **68.96**(5.78)↑ | **98.85**(1.09)↑ | **81.42**(3.23)↑ |
|   + Balanced random (*FSCIL*) | 73.76 | 71.86 | 92.56 | 65.53 | 99.05 | 80.55 |
|   + Full data (*Upper-bound*) | 81.61 | 89.56 | 98.62 | 96.53 | 99.92 | 93.24 |
| DualPrompt [53] | | | | | | |
|   + Random (*Baseline*) | 75.62(0.00) - | 67.85(0.00) - | 93.90(0.00) - | 63.59(0.00) - | 97.89(0.00) - | 79.77(0.00) - |
|   + Entropy [23] | 71.61(4.01)↓ | 66.52(1.33)↓ | 92.70(1.20)↓ | 62.06(0.53)↓ | 98.28(0.39)↓ | 78.23(1.54)↓ |
|   + Margin [41] | 73.92(1.70)↓ | 70.73(2.88)↑ | 92.50(0.40)↓ | 66.87(3.28)↑ | 98.48(0.58)↑ | 80.50(0.73)↑ |
|   + Coreset [45] | 70.38(5.24)↓ | 61.72(6.13)↓ | 89.87(4.03)↓ | 54.37(9.22)↓ | 96.82(1.07)↓ | 74.63(5.14)↓ |
|   + BADGE [7] | 75.07(0.55)↓ | 71.26(3.41)↑ | 94.24(0.34)↑ | 67.03(3.44)↑ | **99.04**(1.15)↑ | 81.32(1.55)↑ |
|   + Typiclust [21] | 76.91(1.29)↑ | 72.98(5.13)↑ | 95.34(1.44)↑ | 68.50(4.91)↑ | 98.77(0.88)↑ | 82.50(2.73)↑ |
|   + ProbCover [62] | 73.88(1.74)↓ | 66.88(0.97)↓ | 94.56(0.66)↑ | 58.18(5.41)↓ | 97.08(0.81)↓ | 78.11(1.66)↓ |
|   + DropQuery [39] | 73.74(1.88)↓ | 71.71(4.86)↓ | 93.93(0.03)↑ | 66.09(2.50)↑ | 98.56(0.63)↑ | 80.80(1.03)↑ |
|   **+ CBS (*Ours*)** | **77.11**(1.49)↑ | **73.50**(5.65)↑ | **95.38**(1.48)↑ | **70.37**(6.47)↑ | 98.71(0.82)↑ | **83.01**(3.24)↑ |
|   + Balanced random (*FSCIL*) | 76.02 | 71.95 | 94.27 | 65.46 | 99.09 | 81.35 |
|   + Full data (*Upper-bound*) | 83.73 | 90.94 | 98.72 | 97.53 | 99.88 | 94.16 |
| LP-DiF [25] | | | | | | |
|   + Random (*Baseline*) | 70.24(0.00) - | 76.01(0.00) - | 93.46(0.00) - | 70.31(0.00) - | 92.24(0.00) - | 80.45(0.00) - |
|   + Entropy [23] | 67.84(2.40)↓ | 68.20(7.81)↓ | 92.95(7.81)↓ | 66.62(3.69)↓ | 89.84(2.40)↓ | 77.09(3.36)↓ |
|   + Margin [41] | 68.41(1.83)↓ | 71.08(4.93)↓ | 93.12(0.34)↓ | 69.84(0.47)↓ | 92.08(0.16)↓ | 78.90(1.55)↓ |
|   + Coreset [45] | 66.21(4.03)↓ | 71.59(4.42)↓ | 92.85(0.61)↓ | 64.66(5.65)↓ | 86.56(5.68)↓ | 76.37(4.08)↓ |
|   + BADGE [7] | 70.05(0.19)↓ | 70.96(5.65)↓ | 93.64(0.18)↑ | 73.25(2.94)↑ | 93.18(0.94)↑ | 80.21(0.24)↓ |
|   + Typiclust [21] | 72.10(1.86)↑ | 73.65(2.36)↓ | 93.71(0.25)↑ | 72.95(2.64)↑ | 93.55(1.31)↑ | 81.19(0.74)↑ |
|   + ProbCover [62] | 66.87(3.37)↓ | 71.55(4.46)↓ | 93.56(0.10)↑ | 64.90(5.41)↓ | 91.13(1.11)↓ | 77.60(2.85)↓ |
|   + DropQuery [39] | 72.07(1.83)↑ | 73.76(2.25)↓ | **93.79**(0.33)↑ | 70.87(0.56)↑ | 93.79(1.55)↑ | 80.85(0.40)↑ |
|   **+ CBS (*Ours*)** | 73.38(3.14)↑ | 76.26(0.25)↑ | 93.74(0.28)↑ | 72.50(2.19)↑ | 94.31(2.07)↑ | 82.03(1.58)↑ |
|   **+ CBS & unlabeld data (*Ours*)** | **75.20**(4.96)↑ | **77.31**(1.30)↑ | 93.77(0.31)↑ | **73.31**(3.00)↑ | **95.25**(3.01)↑ | **82.96**(2.51)↑ |
|   + Balanced random (*FSCIL*) | 72.14 | 76.11 | 93.64 | 70.59 | 94.06 | 81.30 |
|   + Full data (*Upper-bound*) | 80.79 | 82.50 | 95.13 | 81.72 | 97.73 | 87.57 |

CBS. We also conduct forced class-balanced random selection (*i.e.*, few-shot annotations, which is adopted in FSCIL task), and training incremental learner with the fully labeled data as the *upper-bound* reference.

    **Implementation Details.** All experiments are conducted with PyTorch on NVIDIA RTX 2080Ti GPU. We implement ACIL pipeline based on the PyTorch implementations of L2P, DualPrompt, and LP-DiF, respectively. For each CIL method, we incorporate our proposed CBS and compared active learning methods with it. Specifically, for the compared active learning methods, we use them to replace the ClassBalancedSelection function we call in Alg. 1 (line 4). On each dataset, we conduct experiments under the annotation budget size $B \in \{40, 60, 80, \dots, 200\}$ for each session, respectively. Note that our method selects $B$ samples at once for each session, whereas some compared active learning algorithms are based on multiple rounds to selection, labeling, and training. Therefore, for these methods, we maintain their multi-round pipeline and make

them select 20 samples in each round for labeling until the number of selected samples reaches $B$. For more training details, such as the training optimizer, learning rate, batch size, etc., please refer to the *supplementary materials*.

## 4.2 Main Results

    **Comparison with existing active learning methods.** We summarize the experiments results of competing active (AL) methods applying to three CIL methods on five selected datasets under $B = 100$, in Tab. 1. For each CIL method applied with a certain AL method, we report the average accuracy over all incremental sessions on each datasets, with a extra Mean Avg over all datasets. Generally, we can observe that the performance of CIL models trained with samples selected by some SOTA existing AL methods (the rows with grey highlight) is lower than that of random selection. For example, when applying various AL methods to LP-DiF, Entropy, Margin and BADGE underperform random selection

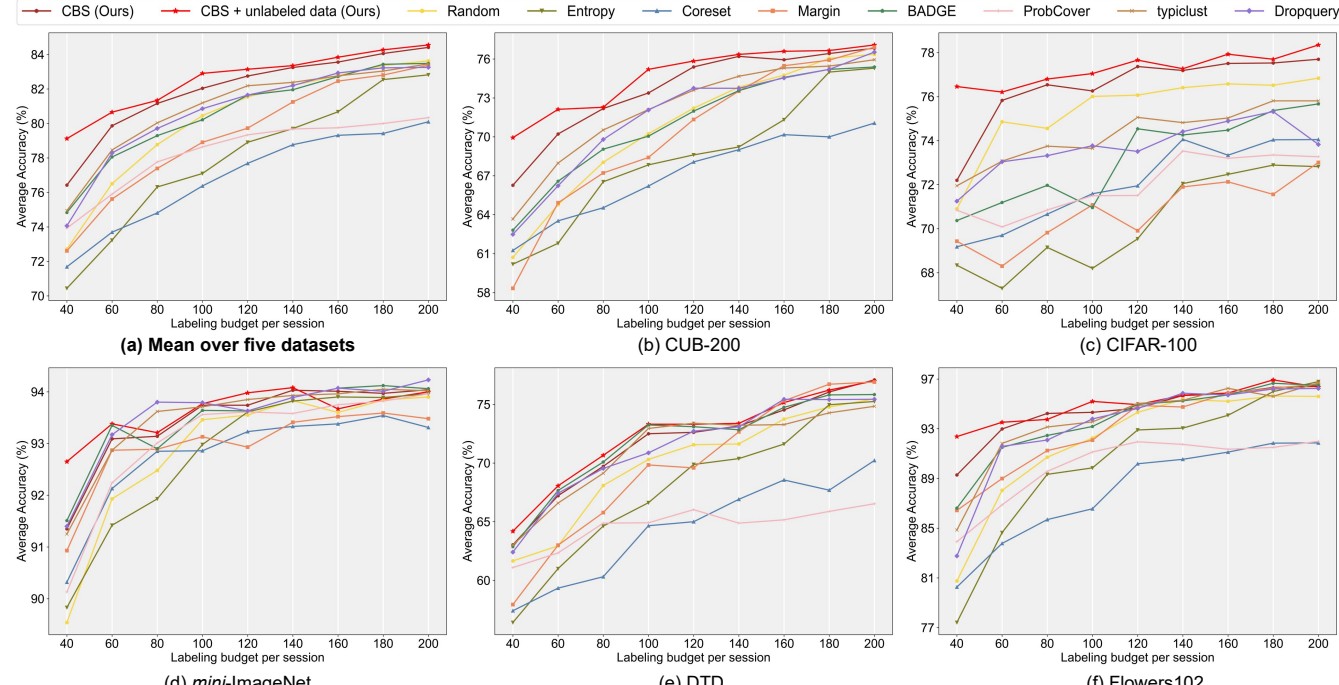

Figure 2: Avg curves of our CBS and comparison with counterparts applied to LP-DiF on five datasets (*i.e.*, (b) to (f)) under various labeling budget $B$. (a) shows the mean Avg curves over five datasets.

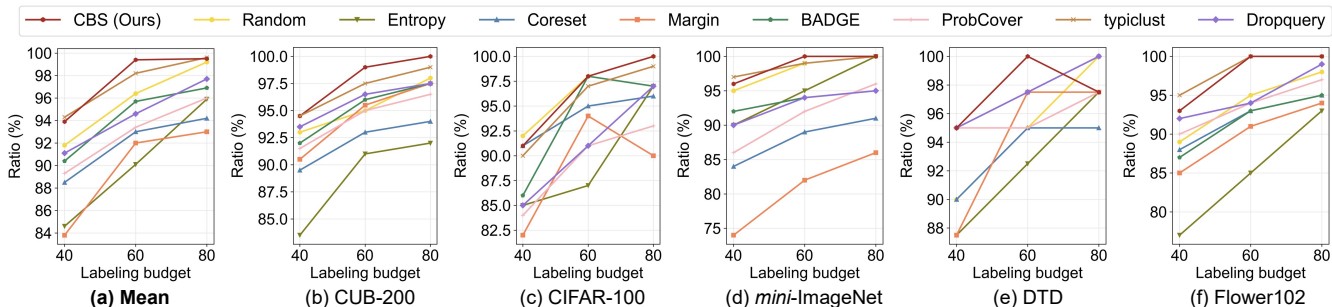

Figure 3: Comparison of "classes discovery ratio" by our CBS and other counterparts applied to LP-DiF on five datasets (*i.e.*, (b) to (f)) under $B \in \{40, 60, 80\}$. (a) shows the mean ratio curves over five datasets.

across all the five datasets. Especially, the performance of all existing AL methods on CIFAR-100 are lower than that of random selection. However, when applied to each CIL method, our proposed CBS outperforms random selection and these existing AL methods on most datasets, and achieve the highest performance in terms of Mean Avg. We believe such results are, to a certain extent, due to the class balance of the samples selected by our method being better compared to random sampling and existing active learning methods. In the supplementary materials, we will report the comparison of class balance of the samples selected by CBS and other counterparts. Moreover, we can also observe that our CBS outperform Balance random for all CIL methods on most datasets, which demonstrate that our CBS can select more informative samples than that of Balance random selection adopted in FSCIL task. In addition, for LP-DiF, our proposed use of unlabeled data for Gaussian distribution estimation shows a more significant improvement over its original LP-DiF for each dataset.

**Comparison under various labeling budget.** Fig. 2 shows the comparison of CBS with counterparts applied to LP-DiF under various labeling budget on five datasets. Each curve corresponds to an active learning method, and each point of each line represents the Avg over all sessions under a specific labeling budget. Generally, one can obtain the following observations: 1) For each dataset, compared to existing SOTA active learning methods and random selection, our proposed CBS achieved the best or comparable performance under any specified labeling budget. Especially under lower labeling budget, *e.g.*, $B = 40$ or $B = 60$, the performance of CBS is significantly higher than other counterparts. 2) For each dataset, our design of using unlabeled data to improve estimating Gaussian distributions further enhanced the performance of the incremental learner, demonstrating the effectiveness of our improvement method. In the supplementary materials, we will demonstrate that the improvement in performance is primarily due to an increase in classification accuracy for the old classes 3) Our CBS and "CBS +

Table 2: Ablation studies of our CBS applied to LP-DiF on CUB-200 under $B = 100$. `KM.` and `GS.` represents K-means and proposed greedy selection strategy respectively. `Ent.`, `CS.` and `BD.` represent Entropy [23], Coreset [45] and BADGE [7], respectively. `ULD.` represents our proposed improvement strategy for estimating distribution by unlabeled data introduced in Sec. 3.3. The 5th row and the 6th row correspond to CBS and CBS & unlabeled data, respectively.

| KM. | Ent. | CS. | BD. | GS. | ULD. | Accuracy in each session (%) ↑ | | | | | | | | | | Avg ↑ |
|---|---|---|---|---|---|---|---|---|---|---|---|---|---|---|---|---|
| | | | | | | 1 | 2 | 3 | 4 | 5 | 6 | 7 | 8 | 9 | 10 | |
| | | | | ✓ | | 86.02 | 73.39 | 75.74 | 73.33 | 74.51 | 71.69 | 68.64 | 66.97 | 65.10 | 66.59 | 72.19 |
| ✓ | ✓ | | | | | 85.24 | 71.80 | 72.96 | 69.63 | 72.73 | 70.07 | 67.09 | 65.62 | 64.10 | 63.95 | 70.31 |
| ✓ | | ✓ | | | | 86.60 | 73.64 | 75.13 | 73.54 | 74.67 | 71.75 | 68.19 | 67.27 | 64.65 | 64.72 | 72.01 |
| ✓ | | | ✓ | | | 86.94 | 72.95 | 74.68 | 73.60 | 74.15 | 72.42 | 68.36 | 66.88 | 64.97 | 65.31 | 72.03 |
| ✓ | | | | ✓ | | **89.71** | 75.69 | 77.52 | 74.60 | 75.80 | 72.86 | 69.17 | 68.49 | 66.73 | 67.22 | 73.38 |
| ✓ | | | | ✓ | ✓ | **89.71** | **75.87** | **79.12** | **76.76** | **77.93** | **74.72** | **71.10** | **70.65** | **68.06** | **68.50** | **75.20** |

unlabeled data" achieves the highest Mean Avg over five datasets under each labeling budget compared to all the counterparts. The above results fully demonstrate the effectiveness of our method.

**Analysis the class balance of selected examples.** In the supplementary materials, we will report the comparison of class distribution of the selected samples in detail. Here, we calculate the ratio of classes corresponding to the samples selected by each active learning method to the total classes of the unlabeled pool, thus to reflect the capability to select class-balanced samples of each active learning method. For concise expression, we name this ratio the "classes discovery ratio". Fig. 3 shows the comparison of "classes discovery ratio" by our CBS and other counterparts applied to LP-DiF on five datasets. We clearly observe that our method can identify a larger proportion of samples compared to other counterparts on most datasets. For example, CBS can find all classes under $B = 60$ and $B = 80$ on *mini*-ImageNet, while the ratio of classes discovered by most SOTA active learning methods is even significantly lower than that of random selection. These results to some extent explains why our CBS outperforms counterparts when the specified number of labeled data is low.

## 4.3 Ablation Studies and Analysis

To explore the effectiveness of each module we proposed, we conducted ablation experiments using LP-DiF as the CIL method, on the CUB-200 dataset under $B = 100$. We report the accuracy on each session as well as the average accuracy over these sessions.

**Effect of K-means.** As introduced in Sec. 3.2, the first step of CBS is to cluster the image features of unlabeled data using k-means, and the second step is greedy selecting samples from each cluster. Here we explore the necessity and effect of performing clustering operations on features. To this end, we conduct an experiment where we skipped the clustering step and directly adopt the designed greedy selection approach to select all the unlabeled features. The comparison results is shown in Tab. 2. Clearly, the performance of the model trained with samples selected directly without clustering (1st row of Tab. 2) is lower in each session compared to the model trained with samples selected from each cluster after performing clustering (5th row of Tab. 2), *i.e.*, 72.19% *vs.*73.38%, which proves that it is meaningful to first cluster all unlabeled features.

**Effect of greedy sampling strategy.** Within each cluster, we use the designed greedy selection strategy to select samples, aiming to efficiently ensure that the distribution of the selected samples is as close as possible to the distribution of the entire cluster, thereby

achieving balanced sampling. A natural question is, if existing active learning methods are adopted to sample within each cluster, would they be able to achieve the same performance as CBS? To this end, we conduct experiments where we replace the designed greedy selection strategy with Entropy [23], Coreset [45], and Badge [7], which respectively represent uncertainty-based methods, diversity-based methods, and hybrid methods in active learning. The experimental results indicate that using our proposed greedy selection approach within each cluster achieves higher performance compared to using these three existing active learning methods within each cluster. This suggests that simply combining clustering with existing active learning methods is still sub-optimal, while the samples selected by our proposed greedy selection approach enable the model to achieve higher performance.

**Effect of using unlabeled data to estimate Gaussian distributions.** When incorporate CBS into LP-DiF [25], we propose using unlabeled data to improve the Gaussian distribution estimated for each old classes, allowing it to sample pseudo features with higher quality. The effects of this strategy are shown in the 6th row of Tab. 2. Obviously, compared to not using unlabeled data (*i.e.*, only using labeled data to estimate the Gaussian distribution, which is proposed in LP-DiF), we can see that our proposed improvement strategy performs better in subsequent incremental sessions. This proves that using unlabeled data can be beneficial for old knowledge replay, and thus enhancing the model's ability to resist catastrophic forgetting.

## 5 CONCLUSION

In this paper, we focus on Active Class-Incremental Learning (ACIL) and empirically discover that existing active learning strategies result in severe class imbalance in the samples selected during each incremental session, which subsequently harms the performance of the incremental learner. To address this, we propose an active selection method named CBS, which considers both the class balance and informativeness of the selected samples to benefit the training of the incremental learner. CBS initially cluster the unlabeled pool into multiple groups via k-means, then uses a greedy selection strategy in each cluster to match the selected samples' distribution closely with the cluster's overall distribution. Our CBS can be plug-and-played into most of the recently popular CIL methods built on pretrained models and employ prompts tunning technique. Extensive experiments on various datasets showcase the superiority compared to existing active learning methods.

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
