# OpenReview forum: "Class Balance Matters to Active Class-Incremental Learning"
_acmmm.org/ACMMM/2024/Conference — MM2024 Poster_

### Official Review · Reviewer_QoQp · 2024-04-29

**Rating:** 4
**Confidence:** 4

**Summary:**

Different form existing few-shot class-incremental Learning, this paper aims to start from a pool of large-scale unlabeled data and then annotate the most informative samples for incremental learning. Specifically, this paper proposes the Class-Balanced Selection (CBS) strategy to ensure the distribution of selected samples closely mirrors the distribution of the entire unlabeled pool. Finally, the proposed method achieve comparable performance to previous works across five diverse datasets.

**Strengths:**

1.	The task of the active class-incremental learning (ACIL) presented in this paper is interesting and meaningful.
2.	A large number of comparative experimental results as well as ablation study results effectively demonstrate the superiority of the proposed method in terms of performance and robustness.

**Limitations:**

1.	The parameters are not clear. For example, in Equation 3, the symbol \tau is not specifically explained. If it is a hyperparameter, relevant ablation experiments are needed.

**Suitability:**

3

---

### Official Review · Reviewer_8Dxr · 2024-05-09

**Rating:** 3
**Confidence:** 3

**Summary:**

The paper introduces a new task namely Active Class-Incremental Learning, in which algorithms should select out the most meaningful data samples from a large pool of unlabeled data for artificial annotation and used for incremental learning. To keep the selected subset informative as well as class-balanced, a class-balanced selection (CBS) method is proposed as a model-agnostic plug-in for general CIL algorithms. To be more specific, CBS clusters unlabeled data into groups and select subsets that reflect the macro feature distribution most by using greedy strategy.

**Strengths:**

1. The presentation of the paper is pretty clear and well-organized. The technical details of proposed CBS is also well-stated. These make the paper easy to follow.
2. The efficient greedy selection design is reasonable, cutting down the potential cost of trying different subset choices.
3. Sufficient and thorough experiments are conducted to evaluate the effectiveness, and relatively good performance can be seen under most settings.

**Limitations:**

1. The motivation and the goal of the proposed new task are somehow weak and ambiguous. Active Learning problem not only focuses on the annotation budget, but also considers the training cost as well as the final results. The demonstration in the paper seems more like to simply put Active Learning method in another scene. As it is shown in a number of incremental learning works, the best upper bound results can be achieved by including all examples into training. However, in the newly proposed ACIL setting, the unchosen unlabeled data are not included in training by semi or unsupervised learning method, which do not rely on artificial annotation. The trade-off goal among annotation budget, computation cost and final result should be carefully discussed when presenting the new task. There should be more illustration about how Active Learning truly effects the incremental learning. If more reasonable explanation about this can be provided, a raise in rating is possible.
2. The proposed distribution-mirroring selection method is not novel. Similar methods have been used in other works like [1].
3. The performance is not consistently good. When testing under mini-ImageNet, the gains are very little or even surpassed by other methods. This needs more discussion.
4. Although the authors announce that the method has the potential to be used in other modals in **Relevance To Conference**, the experiments in the paper restrict to image datasets. I would still suggest more experiment on NLP or Video data, which can support the work's contribution to the MM society.

[1] Feature Distribution Matching by Optimal Transport for Effective and Robust Coreset Selection. Wei X. et al.

**Suitability:**

2

---

### Official Review · Reviewer_KFXG · 2024-05-22

**Rating:** 4
**Confidence:** 3

**Summary:**

The paper "Class Balance Matters to Active Class-Incremental Learning" introduces Active Class-Incremental Learning (ACIL), a method to improve learning new classes with limited annotations by selecting the most informative samples from a pool of unlabeled data. Existing active learning methods often result in class-imbalanced datasets, which degrade incremental learning performance. To address this, the authors propose Class-Balanced Selection (CBS), which clusters features of unlabeled images and uses a greedy selection method to ensure the sampled features' distribution matches the entire unlabeled pool. Experiments on datasets show that CBS outperforms random selection and other active learning approaches.

**Strengths:**

1.Clear and Comprehensible Writing: The paper is well-written, making it easy for readers to understand the proposed ACIL task and the design of the CBS strategy. The authors effectively explain the motivation behind their work, the issues with existing methods, and how their proposed solution addresses these problems.

2.Strong Experimental Results: The experimental results are impressive, demonstrating the effectiveness of the proposed method. The authors provide a detailed ablation study and analysis, which thoroughly examine the contributions of each module within the CBS strategy. This rigorous evaluation not only validates the overall approach but also highlights the significance of individual components.

**Limitations:**

1.Limited Innovative Contribution: The proposed method appears to be primarily a strategy for selecting unsupervised samples using pretrained models. This approach may lack significant innovative contributions, as it builds heavily on existing techniques of leveraging pretrained models for feature extraction and clustering.

2.Implicit Label Information: Although the authors claim that ACIL selects samples from unlabeled images, each session in the CIL setting implicitly contains label information. For example, in the CIFAR-100 CIL setting, each incremental session includes only five or ten classes. This means that the sample selection in ACIL is not entirely unsupervised, as there is an underlying knowledge of the class distribution for each session. This reduces the task's difficulty and diminishes the novelty and contribution of the proposed method.

3.Dependence on Oracle for Labels: The selected images' labels still need to be obtained through an oracle. In many real-world incremental learning scenarios, access to such an oracle may not be feasible, limiting the practical applicability of the proposed method. This reliance on human annotation can be a significant bottleneck in deploying the system in autonomous or resource-constrained environments.

4.Lack of Experiments with Lower Labeling Budgets: The paper does not explore the performance of the proposed method under lower labeling budgets, such as 𝐵 = 10 or 𝐵 = 20. For example, in Few-Shot Class-Incremental Learning (FSCIL), [1] conducts experiments with a 1-shot setting. Demonstrating strong performance under lower labeling budgets would better illustrate the method's superiority and robustness, particularly in scenarios where obtaining labeled data is extremely challenging.

[1]Peng, Can, Kun Zhao, Tianren Wang, Meng Li, and Brian C. Lovell. "Few-shot class-incremental learning from an open-set perspective." In ECCV 2022.

**Suitability:**

2

---

### Official Review · Reviewer_PQCW · 2024-05-23

**Rating:** 2
**Confidence:** 3

**Summary:**

This paper proposes a Class-Balanced Selection (CBS) strategy, which filters the most informative samples. CBS contains class-balanced selection and distribution estimation. The paper presents an Active Class-Incremental Learning task and empirically reveals that class-balanced annotations are crucial for promising incremental learning. And the CBS considers both the class balance and informativeness of the selected samples for benefiting training the incremental learner. The paper incorporates CBS into L2P, DualPrompt, and LP-DiF to represent CIL methods based on pre-trained models employing the prompt tunning technique.

**Strengths:**

(1) The algorithm logic of this paper is clear and easy to understand.

**Limitations:**

(1) Selecting samples via active learning takes much more time.

(2) The paper mentions few-shot class-increment in the abstract and introduction, but has no relevant content in the experiment or other sections.

(3) The authors write the split of data in the supplementary material and I think it can be written in the main text.

**Suitability:**

2

---

### Meta-Review · Area_Chair_X9f9 · 2024-06-29

**Recommendation:** Accept (Poster)
**Confidence:** 5

**Metareview:**

The paper proposes a method to address class imbalance and samples selection in a Few-Shot Class Incremental learning tasks (FSCIL). They proposed method (CBS) aims to exploit the availability of the unlabeled data to select the most informative samples for annotation/training (to reduce the labeling cost) while ensuring the class labels are balanced among the selected samples (to enhance the model performance fairly among all classes).

We thank the authors for reaching out to us expressing their concerns regarding the reviewing process. We have fully examined their concerns.

Strengths:
1. The paper is well-written. The authors have made an effort in writing the paper; conducting the experiments using various benchmarks including images and texts (i.e, the experiments provided in the rebuttal pdf) and responding to the reviewers concerns.


Limitations:
1. I agree with the respected reviewers that the methodology lacks significant novelty and the proposed methods contribute to the fields marginally.
2. The results also look marginals and the proposed method improves the performance slightly compared to SOTA methods.

Conclusion:
The paper is well-written, the efforts in conducting the experiments are visible. Although the methodology is limited I think this work is useful and this work would contribute and benefit researchers in this field.